



**Assimilating Shallow Soil Moisture Observations into Land Models**
**with a Water Budget Constraint**
Bo Dan[1], Xiaogu Zheng[2], Guocan Wu[3*], and Tao Li[4]
[1] National Marine Data and Information Service, Tianjin, China
[2] Key Laboratory of Regional Climate-Environment Research for East Asia, Institute
of Atmospheric Physics, Chinese Academy of Sciences, Beijing, China
[3] College of Global Change and Earth System Science, Beijing Normal University,
Beijing, China
[4] Institute of Statistics, Xi'an University of Finance and Economics, Xi'an, China

*Corresponding author: Guocan Wu
E-mail: gcwu@bnu.edu.cn



**Abstract**
Incorporating observations of shallow soil moisture content into land models is an
important step in assimilating satellite observations of soil moisture content. In this
study, several modifications of an ensemble Kalman filter (EnKF) are proposed for
improving this assimilation. It was found that a forecast error inflation-based
approach improves the soil moisture content in shallow layers, but it can increase the
analysis error in deep layers. To mitigate the problem in deep layers while maintaining
the improvement in shallow layers, a vertical localization-based approach was
introduced in this study. During the data assimilation process, although updating the
forecast state using observations can reduce the analysis error, the water balance
based on the physics in the model could be destroyed. To alleviate the imbalance in
the water budget, a weak water balance constrain filter is adopted.
The proposed weakly constrained EnKF that includes forecast error inflation and
vertical localization was applied to a synthetic experiment and two real data
experiments. The results of the assimilation process suggest that the inflation
approach effectively reduce both the short-lived analysis error and the analysis bias in
shallow layers, while the vertical localization approach avoids increase in analysis
error in deep layers. The weak constraint on the water balance reduces the degree of
the water budget imbalance at the price of a small increase in the analysis error.
**Keywords**
soil moisture, water balance, data assimilation, forecast error inflation, vertical
localization



## 1. Introduction

Soil moisture content is one of the most important variables that affect the water cycle and energy balance through land-atmosphere interactions, especially evaporation and precipitation (Han et al. 2014; Kumar et al. 2014; McColl et al. 2019; Pinnington et al. 2018). Adequate knowledge of the horizontal and vertical distributions of soil moisture could improve weather and climate predictions (Delworth and Manabe 1988; Pielke 2001). Alongside snow cover, soil moisture content is an important component of the meteorological memory of the climate system over land (McColl et al. 2019; Robock et al. 2000; Zhao and Yang 2018). It is also a primary water resource for the terrestrial ecosystem and affects runoff (GUSEV and Novak 2007).

There are several ways to estimate the soil moisture content. Land surface models can provide temporally and spatially continuous estimates of the soil moisture content, but these estimates are limited by the uncertainty in the models' parameters, errors in the forcing data and imperfect physical parameterizations (Bonan 1996; Dai et al. 2003; Dickinson et al. 1993; Oleson et al. 2010; Yang et al. 2009). Compared with the results of models, in-situ observations of the soil moisture content provide more accurate profiles (Bosilovich and Lawford 2002; Dorigo et al. 2011; Robock et al. 2000); however, networks of in-situ observations are usually too sparse to estimate the soil moisture content on a regional scale (Gruber et al. 2018; Loizu et al. 2018). Satellite remote sensing retrievals could provide soil moisture content data on regional scales (Bartalis et al. 2007; Crow et al. 2017; Entekhabi et al. 2010; Kerr et al. 2010; Lu et al. 2015; Njoku et al. 2003), but they are only available for the shallow layer of the soil and the quality is poor in vegetated area (Pinnington et al. 2018; Yang et al. 2009).



A much better approach to improving estimates of soil moisture contents on
regional scales is to constrain land model prediction by assimilating data from
large-scale remote sensing observations of the soil moisture content (Crow and Loon
2006; Crow and Wood 2003; Reichle and Koster 2005). The assimilation of passive
microwave measurements (brightness temperatures) into land surface models can
successfully increases the spatial and temporal coverage by interpolation and
extrapolation to unobserved times and locations, and also provide various land surface
state and flux estimates with reduced uncertainty (De Lannoy and Reichle 2016;
Reichle et al. 2017). Therefore, land surface data assimilation has significantly
improved the utility of surface soil moisture data sets (Crow et al. 2017; Lu et al. 2012;
Lu et al. 2015), and can further improve land surface model initial conditions for
coupled short-term weather prediction (Chen et al. 2014; Santanello et al. 2016; Yang
et al. 2016).
A good estimate of the forecast error covariance matrix is crucial for the
compromise between uncertain observations and imperfect model predictions in data
assimilation (Anderson and Anderson 1999; Miyoshi 2011; Miyoshi et al. 2012; Wang
and Bishop 2003). For the Ensemble Kalman Filter (EnKF) assimilation method, the
forecast error covariance matrix is estimated using the sample covariance matrix of
the ensemble forecasts (Dumedah and Walker 2014; Evensen 1994; Han et al. 2014).
However, it is usually underestimated due to sampling and model errors, which can
eventually results in filter divergence (Anderson and Anderson 1999; Constantinescu
et al. 2007; Yang et al. 2015). To address this problem, it suggests that the forecast
covariance matrix be multiplied by an inflation factor (Dee and Da Silva 1999; Dee et
al. 1999; Li et al. 2012; Zheng 2009). This approach is referred to as inflation, and it
becomes particularly important when the error in the model is large (Bauser et al.



2018; El Gharamti et al. 2019; Liang et al. 2012; Raanes et al. 2019; Wu et al. 2013).
Therefore, it could work well in this situation because of the enormous errors in the
land model.
In this study, a scheme for assimilating synthetic and in-situ shallow observations
of the soil moisture content into land models was developed based on EnKF method,
which can provide a foundation for further satellite data assimilation. For the synthetic
experiment, the CLM 4.0 (Version 4.0 of the Community Land Model, (Lawrence et
al. 2011; Oleson et al. 2010)) was used to generate the "true values" and the CoLM
(Common Land Model, (Dai et al, 2003)) was selected as the forecast operator. The
differences in these two models are referred to the model error in an imperfect land
surface model. The inflation factors are estimated at every observation time step
during the assimilation process by minimizing the -2log-likelihood of the difference
between the forecast and the observation (Liang et al. 2012; Zheng 2009). For
assimilating observations near the surface only, such inflation approach can improve
the estimates of the forecast error statistics in shallow soil layers but may artificially
enlarge the forecast error statistics in deep soil layers. To avoid the possibility of
decreasing the quality of the estimates in deep soil layers, a vertical localization with
weighting of observations is adopted (Janjić et al. 2011). In this approach, a
localization function multiplies the weights on the components of the state vector
according to the distance from state layer to the observation. Moreover, the method
based on the maximum likelihood estimation was proposed to estimate the optimal
localization scale factor. These steps can result in a better prediction of the soil
moisture content in the deep layers.
A major objective of soil moisture data assimilation is to address biases in
models and observations (Koster et al. 2009; Reichle and Koster 2004). In this study,



we only assume that models could be biased, while the soil moisture observations are
assumed to be unbiased. Moreover, the soil moisture observations are restricted in
shallow layer, so there is no observation available to correct the modeled soil moisture
biases in deep layers. If one only removes the bias in shallow layer, it would introduce
error in model dynamics. Therefore in this study, we still use traditional bias-blind
data assimilation framework. Nevertheless, the analysis error is further decomposed to
a short-lived error (random error) and a bias (system error). It demonstrates that the
proposed scheme can reduce the both for soil moisture in shallow layer.

In addition to improve assimilation accuracy, this study also focuses on the

imbalance in the water budget that occurs during the process of assimilating the soil
moisture data. The terrestrial water budget is a key part of the global hydrologic cycle.
A better understanding of the budget can help us to improve our knowledge of
land-atmosphere water exchange and related physical mechanisms and therefore, can
improve our ability to develop models (Pan and Wood 2006). Generally speaking,
analyses do not conserve the water budget due to inconsistencies between predictions
made by models and observations (Li et al. 2012; Pan and Wood 2006; Wei et al. 2010;
Yilmaz et al. 2011; Yilmaz et al. 2012). It is really a problem if the water balance is
violated in a systematic manner (for example, model is biased), which suggests a
trouble in data assimilation. Pan and Wood (2006) proposed a method based on a
strong constraint to reincorporate the water balance. However, this method
redistributes the error among the different terms in the water budget, which could
result in unrealistic estimates (Pan and Wood 2006; Yilmaz et al. 2011).

To overcome this shortcoming, Yilmaz et al. (2011) proposed using a weakly

constrained ensemble Kalman filter (WCEnKF) to reduce the imbalance in the water
budget. In a synthetic study, they concluded that the accuracy of a WCEnKF-based



analysis is close to that of an EnKF-based analysis but the water budget balance
residuals are much smaller than that of an unconstrained filter. Nevertheless, the
observations of the soil moisture content cover the entire column, and a perfect model
was used in their studies. This is not generally true, especially when only satellite
observations are assimilated. In this study, the experiments were further designed to
assimilate surface observations into an imperfect land model.
The structure of this paper is arranged as follows: The data and models used in
this study are described in section 2. The details of the WCEnKF-based method that
incorporates inflation and vertical localization (WCEnKF-Inf-Loc) are provided in
section 3. The experimental designs and evaluations of synthetic and real data
experiments are set in sections 4 and 5. The primary results of the synthetic and real
experiments are given in section 6. The discussion and conclusion comprise sections 7
and 8.

## 2. Models and data


2.1 Study area and in-situ stations
The study area is located in the Mongolian Plateau and comprises approximately
9352 square kilometers between 46 ° and 46.5 ° N and between 106.125 ° and 107 ° E.
The dominant biome is grassland, and no river flows through the area (see Figure 1).
The soil moisture content and related meteorological and hydrological parameters
are monitored by automatic stations maintained by the Coordinated Enhanced
Observing Period Asian Monsoon Project (CEOP AP) (Bosilovich and Lawford 2002;
Lawford et al. 2004). The CEOP AP was launched by the World Climate Research
Programme (WCRP) to develop an integrated global dataset that can be used to
address issues relating to water and energy budget simulations and predictions,





monsoon processes and the prediction of river flows. More details can be found at
http://www.ceop.net. In this study, observations of the soil moisture content from two
stations, the Bayantsagaan Station (BTS 46.7765 ̊N, 107.14228 ̊E) and the
Delgertsgot Station (DGS 46.12731 ̊N, 106.36856 ̊E), were used to validate the
assimilation method. At the BTS, the soil moisture content is measured every half
hour at 3, 10, 20 and 40 cm below the surface. At the DGS, measurements are made at
depths of 3, 10, 40 and 100 cm with the same frequency. Only the observations made
at 6:00 am (same with the overpass time of SMOS satellite) are assimilated, while the
others are used for validation.

2.2 Forcing data

In this study, both synthetic and realistic experiments were conducted to explore

the accuracy of the assimilation schemes. In the synthetic experiments, the
simulations were driven by forcing data (including radiation, wind, pressure, humidity,
precipitation and temperature) from the $0.125 ̊x0.125 ̊$ ERA-Interim dataset (Dee et al.
2011) that had been scaled down to provide a temporal resolution of one hour.

In the realistic experiments, the forcing data comprised hourly measurements of

the wind speed, near-surface air temperature, relative humidity precipitation and
surface pressure at local stations (the BTS and DGS). The downward shortwave and
longwave radiation data used were from model output time series data for the study
area provided by the Japanese Meteorological Agency (Huang et al. 2008).

2.3 Models

The Common Land Model (CoLM) developed by Dai et al. (2003) is a

third-generation land surface model. It combines the best features of three successful
models: the Land Surface Model (LSM, (Bonan 1996)), the Biosphere-Atmosphere
Transfer Scheme (BATS, (Dickinson et al. 1993)) and the 1994 version of the Chinese
Academy of Sciences/Institute of Atmospheric Physics model (IAP94, (Dai et al.
2003)), and is being further developed. The primary characteristics of the model
include 10 unevenly spaced soil layers (see Table 1), one vegetation layer, 5 snow
layers (depending on the snow depth), explicit treatment of the mass of liquid water,
ice and phase changes within the system of the snow and soil, runoff parameterization
following the TOPMODEL concept, a tiled treatment of the sub-grid fraction of the
energy    and    water    budget    balance    (Dai    et    al.    2003)    and    a    canopy
photosynthesis-conductance mode that describes the simultaneous transfer of $CO_2$ and
water vapor into and out of the vegetation. The model parameters include data on the
global   terrain,   elevation,   land   use,   vegetation,   land-water   mask   and   hybrid
FAO/STATSGO soil types from the USGS, which are available at a resolution of 30
arc seconds.
Version 4.0 of the Community Land Model (CLM 4.0) (Lawrence et al. 2011;
Oleson et al. 2010) is the land surface parameterization used with the Community
Atmosphere Model (CAM 4.0) and the Community Climate System Model (CCSM
4.0). The CLM 4.0 includes bio-geophysics, the hydrologic cycle, biogeochemistry
and the dynamic vegetation. CLM 4.0 simulates the bio-geophysical processes in each
sub-grid   unit   independently   and   maintains   its   own   prognostic   variables.   The
parameters used in the CLM4.0 differ from those used in the CoLM. For example, the
soil texture data are derived from the IGBP soil data, and the land use data are derived
from   the   UNH   Transient   Land   Use   and   Land   Cover   Change   Dataset
(http://luh.umd.edu/).
In addition to using different parameters, the two models have different structures.



For example, a model of groundwater-soil water interactions (Niu et al. 2007; Niu et
al. 2005) has been incorporated into the CLM 4.0, while zero water flux at the bottom
of a soil column is assumed in the CoLM. In addition, the CLM 4.0 has the same
vertical discretization scheme as the CoLM (see Table 1), which makes comparing the
results of the two models convenient.

**3. Methods**
3.1 Forecast and observation systems
Using notation similar to that used by Yilmaz et al. (2011), the forecast system
can be written as
$$\mathbf{y}_{n,t}^{f} = \mathbf{M}_{n,t-1}\left(\mathbf{y}_{n,t-1}^{a}\right),$$  (1)
where $t=1, ..., T$ is the time index, $n=1, ..., N$ represents an ensemble member (in this
study, the ensemble size is set to 100), $\mathbf{M}_{n,t-1}$ is a CoLM forced by the $n$-th perturbed
atmospheric forcing, and $\mathbf{y}$ is a state vector containing 126 variables. The superscript
"$f$" and "$a$" specify the forecast and analysis, respectively.
Let $\mathbf{x}$ be the state variables related to the water budget, that comprises of $\mathbf{SM}$ and
$\mathbf{SIC}$ (the soil moisture content and the soil ice content in % at the 10 vertical levels
listed in Table 1), CWC and SWE (the canopy's water content and the snow water
equivalent in kg/m$^2$). In this study, only $\mathbf{x}$ is updated by data assimilation, while the
model propagates changes to the other variables over time.
For the traditional EnKF, the forecast error covariance matrix $\mathbf{P}_t$ is
obtained from the ensemble of their anomalies,
$$\mathbf{P}_t = \frac{1}{N-1}\sum_{n=1}^{N}\left(\mathbf{x}_{n,t}^{f} - \frac{1}{N}\sum_{j=1}^{N}\mathbf{x}_{j,t}^{f}\right)\left(\mathbf{x}_{n,t}^{f} - \frac{1}{N}\sum_{j=1}^{N}\mathbf{x}_{j,t}^{f}\right)^{\mathrm{T}}.$$  (2)
To avoid overestimation of the co-variability between shallow observations and soil





moistures deeper than a threshold layer $s$, the following vertical localization function
with weighting of observations $\boldsymbol{\rho}_s$ (Janjić et al. 2011) will be applied on $\mathbf{P}_t$, i.e.,
$$\boldsymbol{\rho}_s(l) = \exp\left(-\mu_s|d_l - d_o|\right) \tag{3}$$
where $l$ represents for the $l$-level soil layer, $d_l$ and $d_o$ represent the depths of
$l$-level soil layer and observation, respectively. $|d_l - d_o|$ is the Euclidian distance
between the two layers. $\mu_s$ is estimated by minimizing the following mean square
error between vertical localization function Eq (3) and a step function with threshold
layer $s$,
$$M(\mu) = \sum_{l \leq s}\left[\exp\left(-\mu|d_l - d_o|\right) - 1\right]^2 + \sum_{l > s}\left[\exp\left(-\mu|d_l - d_o|\right)\right]^2 \tag{4}$$
The estimated $\mu_s$ is listed in Table 2.

The observations of the soil moisture content are collected at a depth of 3 cm at

6:00 am every day (denoted by $o_t$). The observation system is defined as
$$o_t = \mathbf{h}\mathbf{x}_t + \varepsilon_t, \tag{5}$$
where observational operator $\mathbf{h}$ is a 22-dimensional vector which linearly interpolated
the soil moisture at depths of 2.8 cm and 6.2 cm to depth of 3 cm, $\mathbf{x}_t$ represents the
true values of the state variables related to the water budget at the time step $t$ and $\varepsilon_t$
is the observational error with mean zero and variance $R_t$. Since, the main objective
of this study is for methodology related to linear observational operators. Choosing
the linear interpolation as observational operator is only for convenience.

3.2 Assimilation with water budget constraint

Assimilating data on the soil moisture content usually results in an imbalance in





the water budget. To reduce this imbalance, a weak constraint on the water budget
(Yilmaz et al. 2011) is adopted in this study. The ensemble water budget residual at
time step $t$ can be expressed as
$$r_{n,t} \equiv \beta_{n,t} - \mathbf{c}^{\mathrm{T}} \mathbf{x}_{n,t}^{a} , \qquad (6)$$

where
$$\beta_{n,t} = \mathbf{c}^{\mathrm{T}} \mathbf{x}_{n,t-1}^{a} + Pr_t - Ev_{n,t}^{f} - Rn_{n,t}^{f} , \qquad (7)$$

where $\mathbf{c}$ is a 22-dimensional vector that converts the units to millimeters ($mm$) and
adds up the states in $\mathbf{x}$, the diagnostic variables $Pr_t$, $Ev_{n,t}^{f}$ and $Rn_{n,t}^{f}$ ($mm$) are
scalars specifying the states of the precipitation, evapotranspiration and runoff,
respectively, in each pixel.
The cost function used to estimate the state variables with the weak water budget
constraint (Eq. (6)) is
$$\begin{aligned} J_{n,t}(\mathbf{x}) = & \left(o_t - \mathbf{hx}\right)^{\mathrm{T}} R_t^{-1} \left(o_t - \mathbf{hx}\right) + \left(\mathbf{x} - \mathbf{x}_{n,t}^{f}\right)^{\mathrm{T}} \mathbf{P}_{s,t}^{-1} \left(\mathbf{x} - \mathbf{x}_{n,t}^{f}\right) \\ & + \left(\beta_{n,t} - \mathbf{c}^{\mathrm{T}}\mathbf{x}\right)^{\mathrm{T}} \varphi_t^{-1} \left(\beta_{n,t} - \mathbf{c}^{\mathrm{T}}\mathbf{x}\right) \end{aligned} , \qquad (8)$$

where
$$\varphi_t = \frac{1}{N-1} \sum_{n=1}^{N} \left(\beta_{n,t} - \frac{1}{N}\sum_{j=1}^{N}\beta_{j,t}\right) \times \left(\beta_{n,t} - \frac{1}{N}\sum_{j=1}^{N}\beta_{j,t}\right)^{\mathrm{T}} \qquad (9)$$

is an estimate of the variance of $\beta_{n,t}$ and $\mathbf{P}_{s,t}$ represents a forecast error
covariance matrix defined by
$$\mathbf{P}_{s,t} = \left[\sqrt{\boldsymbol{\lambda}_t}\right]\left[\boldsymbol{\rho}_s\right]\mathbf{P}_t\left[\boldsymbol{\rho}_s\right]\left[\sqrt{\boldsymbol{\lambda}_t}\right]. \qquad (10)$$

where $\mathbf{P}_t$ is defined as Eq. (2); $\left[\boldsymbol{\rho}_s\right]$ is a diagonal matrix which localizes the soil
moisture error (i.e. it is $\boldsymbol{\rho}_s$ defined by Eq. (3) for the soil moisture contents and 1 for
other variables). $\left[\sqrt{\boldsymbol{\lambda}_t}\right]$ is also a diagonal matrix which inflates the forecast soil



moisture error (i.e. it is a scalar $\lambda_t$ for the soil moisture contents and 1 for other
variable). $\lambda_t$ is estimated by minimizing the -2log-likelihood of the difference
between the forecast and the observation (Dee and Da Silva 1999; Liang et al. 2012;
Zheng 2009),
$$-2L_{s,t}(\lambda_t) = \ln\left(\mathbf{h}\mathbf{P}_{s,t}\mathbf{h}^{\mathrm{T}} + R_t\right) + \left(o_t - \mathbf{h}\mathbf{x}_t^f\right)^{\mathrm{T}} \left(\mathbf{h}\mathbf{P}_{s,t}\mathbf{h}^{\mathrm{T}} + R_t\right)^{-1} \left(o_t - \mathbf{h}\mathbf{x}_t^f\right). \qquad (11)$$

The estimated forecast error inflation factor is denoted as $\hat{\lambda}_t$. The perturbed analysis
states of the variables related to water budget can be derived by minimizing Eq. (8),
which has the analytic form
$$\mathbf{x}_{n,t}^a = \mathbf{x}_{n,t}^f + \mathbf{P}_t^a \mathbf{h}^{\mathrm{T}} R_t^{-1} \left(o_t + \varepsilon_{n,t} - \mathbf{h}\mathbf{x}_{n,t}^f\right) + \mathbf{P}_t^a \mathbf{c}\varphi_t^{-1}\left(\beta_{n,t} - \mathbf{c}^{\mathrm{T}}\mathbf{x}_{n,t}^f\right), \qquad (12)$$

where $\varepsilon_{n,t}$ is generated from a normal distribution with mean zero and variance $R_t$,
and its error covariance matrix is
$$\mathbf{P}_t^a = \left(\mathbf{h}^{\mathrm{T}} R_t^{-1}\mathbf{h} + \mathbf{P}_t^{-1} + \mathbf{c}\varphi_t^{-1}\mathbf{c}^{\mathrm{T}}\right)^{-1}, \qquad (13)$$

For estimating the optimal threshold layer, define the -2log-likelihood of the total
difference between the forecasts and the observations,
$$L_s \equiv \sum_{t=1}^{T}(-2L_{s,t}(\hat{\lambda}_t)). \qquad (14)$$

The optimal threshold layer $\hat{s}$ is selected as the smallest number $s$ such that $L_s$ is
the minimum of $\{L_2, L_3, \cdots L_{s+1}\}$. The final analysis state is the selected corresponding
to the optimal threshold layer $\hat{s}$. The complete assimilation procedure is shown in
Figure 2.

**4. Synthetic experiments**
4.1 Experimental design



To investigate the performance of the WCEnKF-based method that incorporates
inflation and vertical local decomposition, synthetic experiments were performed
using the CoLM. Unlike the "perfect model" assumption used in Yilmaz et al. (2011),
the assumptions of this study are accounted for the error in the model, especially the
structural error. Because there were structural differences in the models of the water
cycle (see section 2.3) used in the two models, CLM 4.0 was used to generate the
"true values" (i.e., to perform a reference run) for the synthetic experiments and
CoLM was selected as the forecast operator (i.e., to perform an open-loop run).
Therefore, the CLM 4.0 and the CoLM were both integrated on a $0.125°$ grid (see
Figure 1 for the locations) with a time step of one hour. The assimilation time was set
to 6:00 am every day. The assimilation experiments were conducted with 4 scenarios:
a weakly constrained ensemble Kalman filter (WCEnKF), a weakly constrained
ensemble Kalman filter with inflation (WCEnKF-Inf), a weakly constrained ensemble
Kalman filter with inflation and localization (WCEnKF-Inf-Loc) and an ensemble
Kalman filter with inflation and localization (EnKF-Inf-Loc).
Synthetic observations were obtained by interpolating $\mathbf{SM}_t$ to a depth of 3 cm
and adding noise with a normal distribution ( $N\left(\mu=0, \sigma=0.5\%\right)$ ). The initial state
$\mathbf{x}_0$, was generated by running the CoLM from October 1, 2002 to June 1, 2003. Each
component of the initial state was perturbed using an independent standard Gaussian
random variable times 5% of magnitude of the component. The forcing data were
perturbed in the manner described in Yilmaz et al. (2011). The synthetic experiments
were conducted from June 1, 2003 to October 1, 2003. The state variables for each
pixel were updated independently.


4.2 Validation statistics
4.2.1 Model error and bias
The model errors are defined as the difference between the actual values and the
model's predictions based on true initial values, and the bias is the average of the error
in the model during the relevant period. Let $x_t$ denote the true values of the soil
moisture content at time $t$ for a location and vertical soil layer. $x_t^M$ denotes the model
predicted soil moisture from the true state at the previous time step $t$-1. The model's
bias and error variance for one step can be written as
$$b_M = \frac{1}{a_{ts}} \sum_{t=1}^{a_{ts}} \left( x_t^M - x_t \right), \tag{15}$$

$$v_M = \frac{1}{a_{ts}} \sum_{t=1}^{a_{ts}} \left( x_t^M - x_t \right)^2, \tag{16}$$

where $a_{ts}$ is the number of time steps over which the observations made at 6:00 am
each day are assimilated.
4.2.2 Validation of analysis soil moisture
The true soil moisture content values from 7:00 am to 5:00 am next day are used
to validate analysis states. For a location and vertical soil layer, let $x_{t,h}$ be the true
soil moisture content at hour $h$ on day $t$, and $x_{t,h}^f$ represent the forecasted soil
moisture content at hour $h$ from analysis state $x_t^a$ at 6:00 am on day $t$. The analysis
bias is defined as
$$b_a = \frac{1}{23a_{ts}} \sum_{t=1}^{a_{ts}} \sum_{h=7}^{29} \left( x_{t,h}^f - x_{t,h} \right). \tag{17}$$

The analysis error variance is defined as



$$v_a = \frac{1}{23a_{ts}} \sum_{t=1}^{a_{ts}} \sum_{h=7}^{29} \left( x_{t,h}^f - x_{t,h} \right)^2$$
$$= \frac{1}{23a_{ts}} \sum_{t=1}^{a_{ts}} \sum_{h=7}^{29} \left( x_{t,h}^f - x_{t,h} - b_a \right)^2 + b_a^2$$ .
(18)

(See Appendix A for the proof)
4.2.3 Water balance
Following Yilmaz (2011), the water budget imbalance at location is evaluated
using the water balance residual,
$$R = \frac{1}{Na_{ts}} \sum_{t=1}^{a_{ts}} \sum_{n=1}^{N} r_{n,t}$$ .
(19)


## 352 5. Real data experiments

In addition to the synthetic experiments, experiments in which the soil moisture
content observed at the DGS and BTS were assimilated into the CoLM were
conducted. In these experiments, the value of soil moisture was extracted from the
output of the Global Land Data Assimilation (GLDAS)/CLM 2.0 model, which has
been integrated continuously since 1979 (Rodell et al. 2004), and used to initialize the
CoLM. Then, the model was run from October 1, 2002 to June 1, 2003. The states
obtained at the end of the period were used as the initial states. In these experiments,
the initial perturbation scheme, observation error variance, assimilation frequency and
assimilation time were adopted from the synthetic experiments. The forcing data sets
were in-situ observed; they were much more accurate than the ERA-Interim reanalysis
data and were not perturbed.
In the realistic assimilation experiments, the truth is not known. Observations of
the soil moisture content at hours not assimilated (7:00 am to 5:00 am next day) were
used for validation. The analysis bias is estimated as



$$B_a = \frac{1}{23a_{ts}} \sum_{t=1}^{a_{ts}} \sum_{h=7}^{29} \left( \mathbf{hx}_{t,h}^f - o_{t,h} \right)$$
$$\approx \frac{1}{23a_{ts}} \sum_{t=1}^{a_{ts}} \sum_{h=7}^{29} \left( \mathbf{h} \left( \mathbf{x}_{t,h}^f - \mathbf{x}_{t,h} \right) \right) \quad , \tag{20}$$

and the analysis error variance is estimated as
$$V_a = \frac{1}{23a_{ts}} \sum_{t=1}^{a_{ts}} \sum_{h=7}^{29} \left( \mathbf{hx}_{t,h}^f - o_{t,h} \right)^2$$
$$\approx \frac{1}{23a_{ts}} \sum_{t=1}^{a_{ts}} \sum_{h=7}^{29} \left( \mathbf{h} \left( \mathbf{x}_{t,h}^f - \mathbf{x}_{t,h} \right) - B_a \right)^2 + B_a^2 + C \tag{21}$$

where C is a constant which is independent of prediction schemes (See Appendix B
for the proof)
Finally, the water balance residual is defined similar to Eq. (19).

**6. Results**
In the synthetic experiments, the magnitudes of the model's bias and error were
calculated using Eqs (15) and (16), respectively, and are shown in Figure 3. It shows
that the model's bias was almost negative from Figure 3a. The negative bias in the
surface layer was the result of a combination of a lower surface roughness and a larger
leaf area index in the CoLM; these values led to more soil evaporation and more
canopy interception and could result in a smaller amount of water infiltrating the soil
than the amount modeled using the CLM 4.0. In the CoLM, the porosity of each layer
was less than it was in the CLM 4.0, which retained less water and contributed to the
negative bias of the upper 9 layers. However, the magnitude of the bias increased to 2%
in the bottom layer. The significant difference between the two models at the bottom
layer could be ascribed to their different boundary conditions. Interactions between
the soil moisture content and the ground water at the bottom of the soil column were
modeled in the CLM 4.0 (Oleson et al. 2010) but not in the CoLM. The error in each
model (Figure 3b) fluctuated in a manner similar to that of the model's bias. Unbiased
observations are necessary for correcting bias in a model, which is not possible in
many realistic applications, especially in assimilating remote sensing retrievals. Since
satellite observations of the soil moisture content of deep layers are unavailable, only
removing the bias in shallow layers would introduce error in model dynamics.

6.1 Forecast error inflation and vertical localization

In the synthetic experiments, the study domain comprised 40 pixels. Each point in

the grid-scale threshold layer, the localization scale factor $\mu_s$, was determined
independently. Therefore, totally 9 sets of experiments with different localization
scale factor (see Table 2) were conducted separately. Among these experiments, the
"optimal" case for each pixel was defined as the case in which the column averaged
analysis error (Eq. (18)) was minimized (shown in Figure 4). According to Figure 4a,
the corresponding threshold layer $s$ of $\mu_s$ was generally between 5 and 6 in both
cases, which could be ascribed to the homogeneous soil texture and land cover. In the
WCEnKF-Inf-Loc, there were 19 pixels in which the threshold layers were "optimal,"
and the layers selected in the other pixels were suboptimal (most were roughly one
layer away from the "optimal" case). As shown in Figure 4b, the spatial average of the
root analysis error variance (Eq. (18)) of the WCEnKF-Inf-Loc (4.09%) was
comparable with the optimal value (3.84%) even though $s$ was not selected on the
basis of minimizing the analysis error.

The spatial average of the root analysis error variance in each layer in the

schemes with (WCEnKF-Inf-Loc and WCEnKF-Inf) and without (WCEnKF)
inflation are displayed in Figure 5a. Above 62.0 cm, the analysis errors of the schemes
without inflation were substantially larger than those of the schemes with inflation for



the synthetic experiments. This suggested that inflation provided a better estimate in
the layers close to observation. When no inflation was performed, the accuracy of the
soil moisture content was barely improved over that of the simulation case (shown in
Figure 5a).

By comparing the schemes with (WCEnKF-Inf-Loc) and without (WCEnKF-Inf)

vertical localization, the impact of this approach on the assimilation accuracy in each
layer is shown in Figure 5a. Because the threshold layer of the localization function
$\rho_s$ was layer 6 (36.6 cm) for 28 of the pixels (see Figure 4a), the spatial average of
root analysis error variance of the results of the WCEnKF-Inf-Loc is almost identical
to that of the results of the WCEnKF-Inf for depths above 36.6 cm. In contrast,
inflation increased the analysis error in the soil moisture content of the deep layers in
the WCEnKF-Inf. In this model, the sample error covariances of the moisture contents
of shallow and deep soil were inflated by a factor greater than 6 (the average inflation
factor was 6.25). This could lead to larger assimilation errors for deep soil moisture
profiles in the WCEnKF-Inf. Therefore, inflation should be used with vertical
localization to reduce the spurious covariance resulting from the covariance
inflation-based approach.

As it was in the synthetic experiments, vertical localization (WCEnKF-Inf-Loc)

was helpful in avoiding erroneous estimates of the soil moisture contents at lower
levels (in the WCEnKF-Inf). A comparison of the analysis error at a depth of 3 cm
(i.e., the depth of the assimilated observations was 3 cm) in the models with
(WCEnKF-Inf and WCEnKF-Inf-Loc) and without (WCEnKF) inflation showed that
the inflation technique significantly reduces the analysis error at the depth at which
observations are made.

In the real data experiments, the spatial averages of root analysis error variance



in each layer (Eq. (21)) are shown in Figures 6a and 7a. To validate the effect of the
vertical localization, the results of the "optimal" (based on the minimum analysis error
at the four observation sites) and WCEnKF-Inf-Loc were compared. In the
experiments using the data from the DGS, the threshold, $s$, was set to layer 2 (2.8 cm)
for the "optimal" case and layer 5 (21.2 cm) for the WCEnKF-Inf-Loc. However, the
analysis error in the two models at each layer in which observations were made
remained comparable. In the experiments using the data from the BTS, the value of $s$
was set to 3 (6.2 cm) in both models, which resulted in equivalent analysis errors.

Unlike the truth at all model depths are available in the synthetic experiments,

the observations only available at the four depths for the two stations, which did not
cover the all model layers. Therefore, the analysis error in layers deeper than the
observation could not be checked.

6.2 The water budget constraint

In the synthetic experiment, the weak constraint on the water budget reduced the

water balance residual significantly in each pixel and the results are shown in Figure 8.
It shows that, the water balance residuals for the assimilation scheme with water
budget constraint are smaller than those without water budget constraint. The forecast
error covariance matrix inflation can lead to the increase of water balance residual,
while the vertical localization technique (i.e. WCEnKF-Inf-Loc scheme) can restrict it
in a rational range. In the WCEnKF-Inf-Loc, the spatial average of the water balance
residual was 0.0742 mm, which was much less than that of the EnKF-Inf-Loc (0.2259
mm). The spread of the water balance residual was also smaller in the
WCEnKF-Inf-Loc, which signals a more stable water balance budget. Therefore, the
weak constraint on the water budget resulted in an assimilation accuracy that was



comparable to that of unconstrained filters but had a much smaller water budget
residual, which is consistent with the results of previous studies (Yilmaz et al. 2011;
Yilmaz et al. 2012).
To investigate the role of the water budget constraint in the assimilation process
in the synthetic experiment, the spatial averaged root analysis error variance (Eq. (18))
of the schemes with (WCEnKF-Inf-Loc) and without (EnKF-Inf-Loc) the water
budget constraint were compared. In the EnKF-Inf-Loc, the threshold layers were
adopted from the WCEnKF-Inf-Loc. According to Figure 5a, the spatial averaged root
analysis error variances of the two models were almost identical (1.83% for the
WCEnKF-Inf-Loc and 2.00% for the EnKF-Inf-Loc) in the layers that were shallower
than 21.2 cm. However, for the layers that were deeper than 36.6 cm, the average
RSME of the EnKF-Inf-Loc (4.95%) was less than that of the WCEnKF-Inf-Loc
(5.87%). This could be the compensation for the reduction in the water balance
residual.
In the real data experiments, consistent reductions in the water budget residual
were obtained from the different experiments. The water balance residuals (Eq. (19))
in the EnKF-Inf-Loc at the DGS and BTS were 0.1545 mm and 0.1792 mm,
respectively. In contrast, the residuals were reduced to 0.0386 mm and 0.0131 mm,
respectively, at the two stations in the WCEnKF-Inf-Loc, which supports the
robustness of the weak constraint on the water budget.

**7. Discussion**
**7.1** Covariance inflation and vertical localization
In this study, the cost function used to estimate the state variables with the weak
water budget constraint (Eq. (8)) consists of three parts, which are related with



observations, model forecasts and water residual (Yilmaz et al. 2012). It is represented
as a summation of three scalars, no matter how many observations are assimilated.
Therefore, inflating of one scalar (e.g., model forecasts) seems to have the similar
impact as deflating another one (e.g., water residual), particularly the weights
associated in this problem can be shown as function of the ratio of these three scalars.
Specifically, inflation of forecast error covariance has somewhat similar impact with
deflation the water balance residual covariance. Accordingly, it is plain obvious that
the water balance residual of the scheme WCEnKF-Inf is larger than that of the
scheme WCEnKF.According to Figures 5a-7a, the covariance inflation improved the
estimates of the soil moisture content in the shallow layers independently of whether
vertical localization was used. This is primarily because the observation operator, $\mathbf{h}$, is
the linear operator that was used to interpolate the soil moisture content at depths of
2.8 cm and 6.2 cm to a depth of 3 cm. Then, the likelihood function for the inflation
factor (Eq. (11)) depends only on the observations and predictions of the soil moisture
content in the $2^{nd}$ and $3^{rd}$ layers. The mean value of the inflation factor is 6.25 for
WCEnKF-Inf, indicating that the initial forecast spread is not large enough. This leads
to an improvement in the forecast error statistics in the shallow layers, and to further
improvements in the soil moisture contents of those layers. However, the soil moisture
contents of the deep layers are not directly related to the inflation factor. Inflating the
forecast errors in the deep layers leads to an overestimation of the corresponding
forecast error covariance, and could lead to larger analysis errors in the deep layers
(see WCEnKF-Inf in Figure 5a). Therefore in this study, the vertical localization
approach was developed to prevent soil moisture over fitting for deep layers. Using all
observations for shreshould $s$ is only for model selection (from the 10 layers), not for
fitting parameter.When vertical localization is used, the soil moisture contents of the





deep layers are not significantly updated. Consequently, larger errors are avoided in
the deep layers (see WCEnKF-Inf-Loc in Figure 5a).

Comparing to traditional EnKF without inflation and localization, although

mainly the soil moisture contents of layers above the threshold layer (usually the 5$^{th}$ or
6$^{th}$ layer) were updated at each time step during the assimilation process when the
WCEnKF-Inf-Loc was used, Figure 5a shows that the soil moisture contents of the
layers below the threshold layer, especially the 6$^{th}$ and 7$^{th}$ layers, are also improved.
This may be because the model propagates changes in the shallow layers downward,
adjusting the soil moisture contents of the deep layers. Because the soil moisture
content of layers above the threshold layer was improved during the previous time
step, this process results in better predictions of the soil moisture contents of layers
below the threshold layer, and therefore, reduces the analysis error in layers below the
threshold layer.

7.2 Bias correction

Geophysical models are never perfect and usually produce estimates with biases

that vary in time and in space (Reichle 2008). Therefore bias correction is important
for assimilating data into models. The model bias can be removed when all model
variables are observed, such as the case studied by Yilmaz et al (2011). However in
this study only soil moisture in shallow layers can be observed (in order to mimic the
satellite observation). There is no observation available to correct the bias of soil
moistures in deeper layers. If only remove the bias in shallow layers, it would
introduce error in model dynamics. Therefore in this study, we still use traditional
(bias-blind) data assimilation framework.

However in the present study, the analysis error variance was decomposed to a



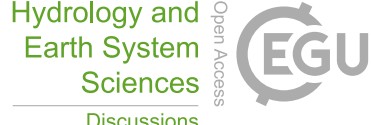

short-lived component (Figures 5b-7b) and a bias component (Figures 5c-7c) for the
synthetic experiment and the two real data experiments, respectively. It shows that for
our proposed bias-blind data assimilation scheme (WCEnKF-Inf-Loc), both
short-lived errors and biases reduce in the layers close to observation, while maintain
the similar levels for the deeper layers. The covariance inflation can play an important
role in bias reduction. Bias can only be seen during whole assimilation period. At an
instant time, bias and error are mixed. For the traditional EnKF, the forecast error
covariance matrix obtained from the ensemble of their anomalies (Eq. (2)) mainly
represents short-lived error, so it has to be inflated to include error related to bias.

There are other bias estimation approaches in data assimilation. For example,
treading bias as model variables and estimate in assimilation (De Lannoy et al. 2007;
Dee and Da Silva 1997; Dee and Da Silva 1998), adjusting the state variable of the
forecast model not only their covariance matrix in each forecast step (Zhang et al.
2015; Zhang et al. 2014), addressing the biases in the model and observations by
rescaling their cumulative distribution functions (Koster et al. 2009; Reichle and
Koster 2004). The scheme proposed here can provide a base line to validate the
efficacy of these approaches and could be further improved after these bias
corrections.

**8. Conclusions**

In this study, observations of the soil moisture content at a depth of 3 cm were
assimilated using an ensemble Kalman filter with three improvements. Firstly, an
adaptive forecast error inflation based on maximum-likelihood estimation was
adopted to reduce the analysis error. This study supports the idea that the proper form
of the forecast error covariance matrix is crucial for reducing the analysis error near





the layers in which observations are made. Secondly, an adequate vertical localization
for the ensemble-based filter was proposed associated with the forecast error
covariance inflation, to avoid misestimates of the soil moisture contents of deep layers.
Lastly, a constraint on the water balance was used in this study to reduce the water
budget residual substantially without significantly changing the assimilation accuracy.
The experiment results of synthetic study and real data show that the
WCEnKF-Inf-Loc assimilation scheme can reduce both the short-lived analysis error
and the analysis bias in the shallow layers, which also lead to a rational water budget
residual.
The work presented in this paper may have some limitations. For example, the
iterations involved in the optimization process reduce the computational efficiency,
and the study area was homogeneous grassland without a compound type of land
cover. Because the accuracy of the microwave soil moisture content is significantly
affected by the land cover type (Dorigo et al. 2010), it is necessary to perform more
experiments using other regions.
In the near future, we plan to validate the major conclusions under different soil
conditions and land cover types. Vertical localization, which uses adjacent
observations, should also be tested in future work. More detailed analyses of the bias
correction for assimilating remote sensing retrievals should be performed. The
response of the analytic soil moisture content to weather predictions also needs to be
investigated. Completing these studies should improve the state of research into
land-atmosphere interactions.





**Data availability** The soil moisture observation and hourly measurements of forcing
data are available at http://www.ceop.net. The ERA-interim forcing data used in the
synthetic experiments is obtained from https://apps.ecmwf.int/datasets. The
downward shortwave and longwave radiation data used in the realistic experiments
are provided by the Japanese Meteorological Agency at https://www.jma.go.jp/en.

**Author Contributions** BD performed the simulations and assimilations. XZ designed
the research. GW analyzed the results. TL collected and preprocessed the data. GW
and XZ prepared the manuscript with contributions from all co-authors.

**Conflicts of Interest** The authors declare that they have no conflict of interest.

**Acknowledgement** This study was funded by the National Basic Research Program
of China (2015CB953703), the National Key R&D Program of China
(2017YFA0603601) and the National Natural Science Foundation of China
(41405098 and 41705086). We would like to thank the Editor and three anonymous
reviewers for their insightful comments in improving the manuscript. We also thank
Drs. Yongjiu Dai and Qingyun Duan for their help in land surface model.

**Appendix A. Proof of Eq. (18)**
For a location and vertical soil layer, the analysis error variance in the synthetic
experiment is defined as





$$v_a = \frac{1}{23a_{ts}} \sum_{t=1}^{a_{ts}} \sum_{h=7}^{29} \left( x_{t,h}^f - x_{t,h} \right)^2$$

$$= \frac{1}{23a_{ts}} \sum_{t=1}^{a_{ts}} \sum_{h=7}^{29} \left( x_{t,h}^f - x_{t,h} - b_a + b_a \right)^2 \qquad \text{(A1)}$$

$$= \frac{1}{23a_{ts}} \sum_{t=1}^{a_{ts}} \sum_{h=7}^{29} \left( x_{t,h}^f - x_{t,h} - b_a \right)^2 + b_a^2 + \frac{2b_a}{23a_{ts}} \sum_{t=1}^{a_{ts}} \sum_{h=7}^{29} \left( x_{t,h}^f - x_{t,h} - b_a \right)$$

From the definition of analysis bias (Eq. (17)), the last term on the right hand side of
is zero, so Eq. (18) is proved.

**Appendix B. Proof of Eqs. (20)-(21)**

Since

$$B_a = \frac{1}{23a_{ts}} \sum_{t=1}^{a_{ts}} \sum_{h=7}^{29} \left( \mathbf{hx}_{t,h}^f - o_{t,h} \right)$$

$$= \frac{1}{23a_{ts}} \sum_{t=1}^{a_{ts}} \sum_{h=7}^{29} \left( \mathbf{hx}_{t,h}^f - \mathbf{h}x_{t,h} - \varepsilon_{t,h} \right) \qquad \text{(B1)}$$

$$= \frac{1}{23a_{ts}} \sum_{t=1}^{a_{ts}} \sum_{h=7}^{29} \left( \mathbf{h}\left( \mathbf{x}_{t,h}^f - \mathbf{x}_{t,h} \right) \right) - \frac{1}{23a_{ts}} \sum_{t=1}^{a_{ts}} \sum_{h=7}^{29} \varepsilon_{t,h}$$

The second term of the right-hand side of Eq. (B1) is approximate zero, because the
observation error $\varepsilon_{t,h}$ has zero mean. Therefore Eq. (20) holds.

Since

$$V_a = \frac{1}{23a_{ts}} \sum_{t=1}^{a_{ts}} \sum_{h=7}^{29} \left( \mathbf{hx}_{t,h}^f - o_{t,h} \right)^2$$

$$= \frac{1}{23a_{ts}} \sum_{t=1}^{a_{ts}} \sum_{h=7}^{29} \left( \mathbf{hx}_{t,h}^f - \left( \mathbf{h}x_{t,h} + \varepsilon_{t,h} \right) - B_a + B_a \right)^2$$

$$= \frac{1}{23a_{ts}} \sum_{t=1}^{a_{ts}} \sum_{h=7}^{29} \left( \mathbf{h}\left( \mathbf{x}_{t,h}^f - x_{t,h} \right) - B_a \right)^2 + B_a^2 + \frac{1}{23a_{ts}} \sum_{t=1}^{a_{ts}} \sum_{h=7}^{29} \varepsilon_{t,h}^2 \qquad \text{(B2)}$$

$$+ \frac{1}{23a_{ts}} \sum_{t=1}^{a_{ts}} \sum_{h=7}^{29} \left( \mathbf{h}\left( \mathbf{x}_{t,h}^f - x_{t,h} \right) - B_a \right) B_a$$

$$+ \frac{1}{23a_{ts}} \sum_{t=1}^{a_{ts}} \sum_{h=7}^{29} \left[ \mathbf{h}\left( \mathbf{x}_{t,h}^f - x_{t,h} \right) - B_a \right] \varepsilon_{t,h} + \frac{B_a}{23a_{ts}} \sum_{t=1}^{a_{ts}} \sum_{h=7}^{29} \varepsilon_{t,h}$$





The third term of the right-hand side Eq. (B2) is denoted as C, it is determined by all
the true values and observations, but not related to any prediction scheme. By the
definition of analysis bias $B_a$ (Eq. 20), the fourth term of the right-hand side Eq. (B1)
is approximate zero; since the observation error $\varepsilon_{t,h}$ has zero mean and is
statistically independent of the forecast error $\mathbf{h}\left(\mathbf{x}_{t,h}^{f} - x_{t,h}\right)$, the fifth and sixth terms
of the right-hand side Eq. (B1) are approximate zero too. Therefore, Eq. (21) holds.





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



**Figure captions**

Figure 1. The topography and river distribution (left plot) and the geographical location of the synthetic study area and the two application stations, the DGS and the BTS (right plot).

Figure 2. The assimilation procedure and localization scale factor estimation in the experiments. All of the equations are in accordance with that described in the text.

Figure 3. The areal average of the model's bias (a) and error (b) for one step in the soil moisture content between the CoLM and the CLM 4.0. The horizontal axis represents the layer depth.

Figure 4. The threshold layers and analysis error for each pixel in the synthetic experiment. Graph (a) illustrates the optimal and WCEnKF-Inf-Loc threshold layers of each pixel. Graph (b) shows the column RSME of each pixel in different schemes with water balance constraint (Optimal, WCEnKF-Inf-Loc, WCEnKF-Inf and WCEnKF). The horizontal axes of (a) and (b) represent the 40 pixels in the study domain.

Figure 5. The assimilation results in each layer for an ensemble Kalman filter with forecast error inflation and vertical localization (EnKF-Inf-Loc), a weakly constrained ensemble Kalman filter with forecast error inflation and vertical localization (WCEnKF-Inf-Loc), a weakly constrained ensemble Kalman filter with forecast error inflation (WCEnKF-Inf), a weakly constrained ensemble Kalman filter (WCEnKF), traditional assimilation (EnKF) and an open-loop simulation. Graphic (a) is for spatial





averaged analysis error of the soil moisture content, (b) is for the short-lived error and
(c) is for the analysis bias.

Figure 6. The assimilation results in each observation layer for an ensemble Kalman
filter with forecast error inflation and vertical localization (EnKF-Inf-Loc), a weakly
constrained ensemble Kalman filter with forecast error inflation and vertical
localization (WCEnKF-Inf-Loc), a weakly constrained ensemble Kalman filter with
forecast error inflation (WCEnKF-Inf), a weakly constrained ensemble Kalman filter
(WCEnKF), traditional assimilation (EnKF) and an open-loop simulation. Graphic (a)
is for spatial averaged analysis error of the soil moisture content, (b) is for the
short-lived error and (c) is for the analysis bias.

Figure 7. Same as Figure 6, but for BTS station.

Figure 8. The box plot of the water balance residual in all 40 pixels for the
EnKF-Inf-Loc, WCEnKF-Inf-Loc,WCEnKF-Inf, WCEnKF and EnKF assimilation
schemes.




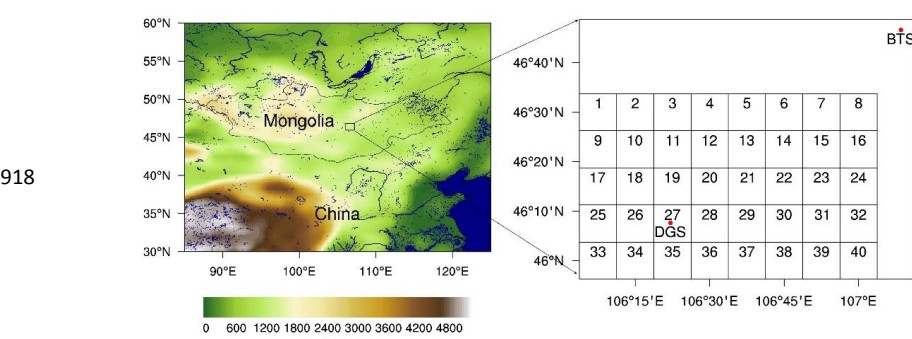


Figure 1. The topography and river distribution (left plot) and the geographical
location of the synthetic study area and the two application stations, the DGS and the
BTS (right plot).









Figure 2. The assimilation procedure and localization scale factor estimation in the
experiments. All of the equations are in accordance with that described in the text.






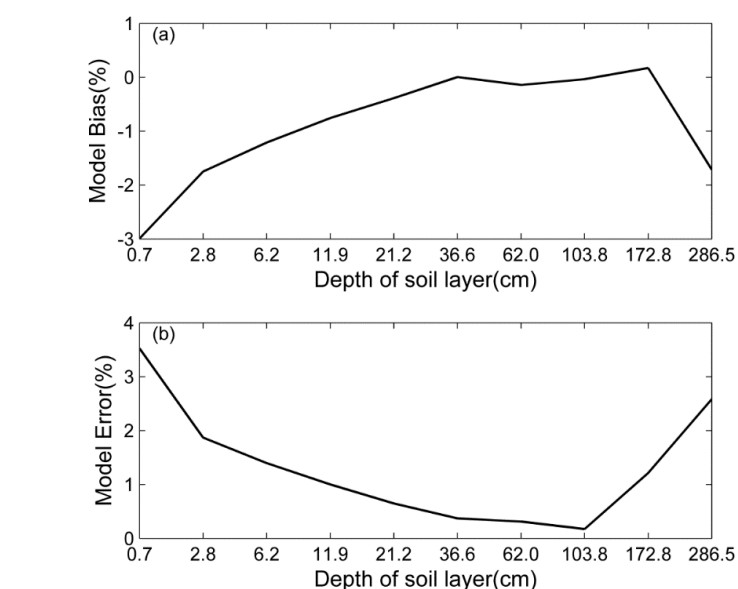

Figure 3. The areal average of the model's bias (a) and error (b) for one step in the soil
moisture content between the CoLM and the CLM 4.0. The horizontal axis represents
the layer depth.




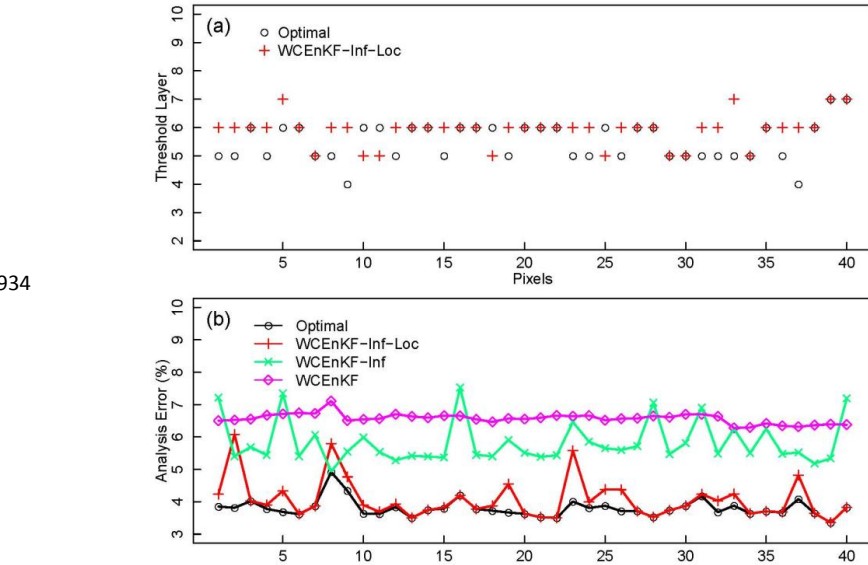

Figure 4. The threshold layers and analysis error for each pixel in the synthetic
experiment. Graph (a) illustrates the optimal and WCEnKF-Inf-Loc threshold layers
of each pixel. Graph (b) shows the column RSME of each pixel in different schemes
with water balance constraint (Optimal, WCEnKF-Inf-Loc, WCEnKF-Inf and
WCEnKF). The horizontal axes of (a) and (b) represent the 40 pixels in the study
domain.



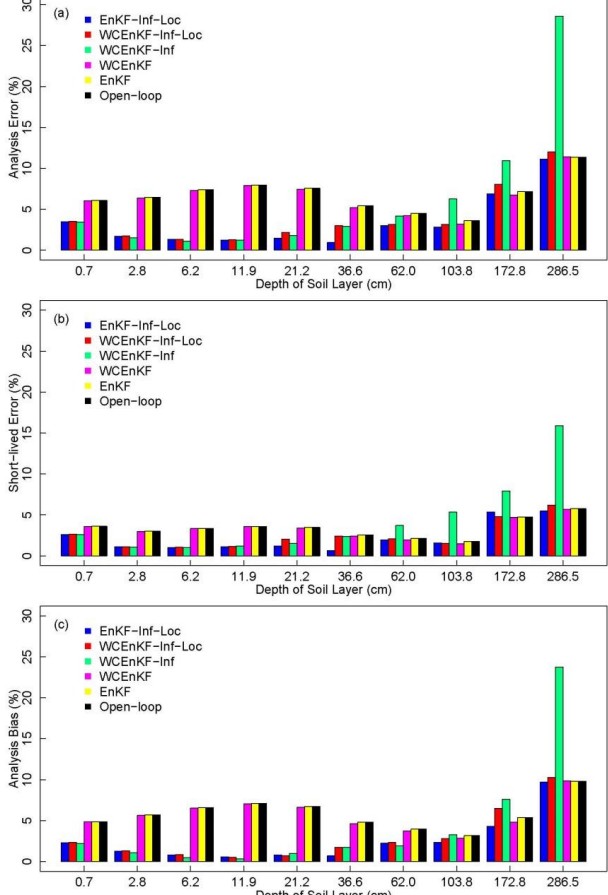

Figure 5. The assimilation results in each layer for an ensemble Kalman filter with
forecast error inflation and vertical localization (EnKF-Inf-Loc), a weakly constrained
ensemble Kalman filter with forecast error inflation and vertical localization
(WCEnKF-Inf-Loc), a weakly constrained ensemble Kalman filter with forecast error
inflation (WCEnKF-Inf), a weakly constrained ensemble Kalman filter (WCEnKF),
traditional assimilation (EnKF) and an open-loop simulation. Graphic (a) is for spatial
averaged analysis error of the soil moisture content, (b) is for the short-lived error and
(c) is for the analysis bias.



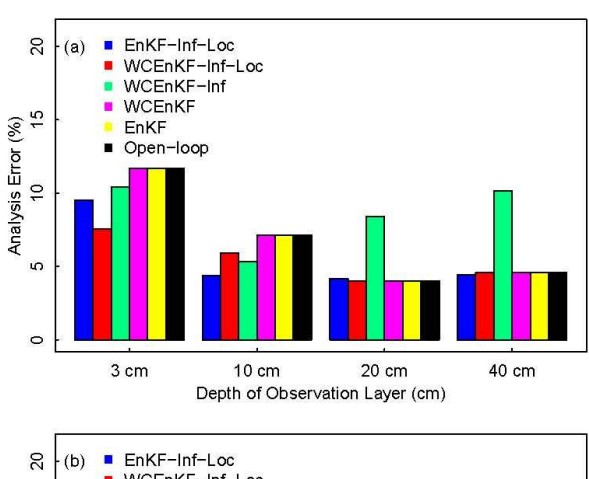


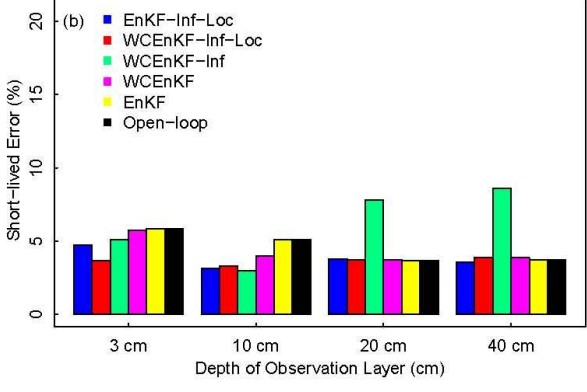

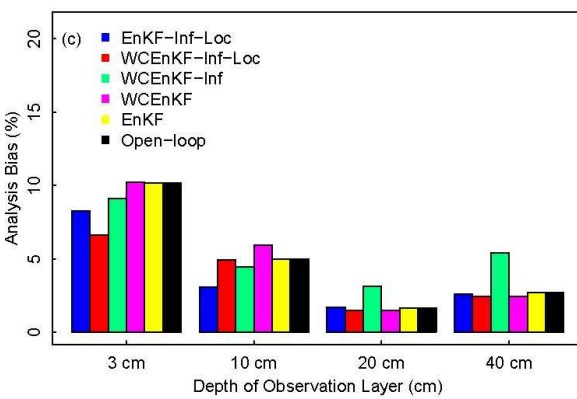

Figure 6. The assimilation results in each observation layer for an ensemble Kalman
filter with forecast error inflation and vertical localization (EnKF-Inf-Loc), a weakly
constrained ensemble Kalman filter with forecast error inflation and vertical
localization (WCEnKF-Inf-Loc), a weakly constrained ensemble Kalman filter with



forecast error inflation (WCEnKF-Inf), a weakly constrained ensemble Kalman filter
(WCEnKF), traditional assimilation (EnKF) and an open-loop simulation. Graphic (a)
is for spatial averaged analysis error of the soil moisture content, (b) is for the
short-lived error and (c) is for the analysis bias.





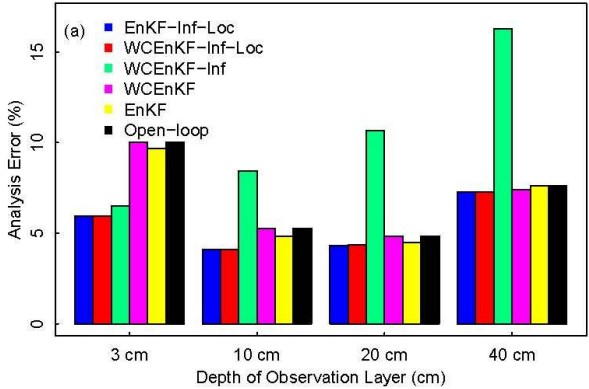


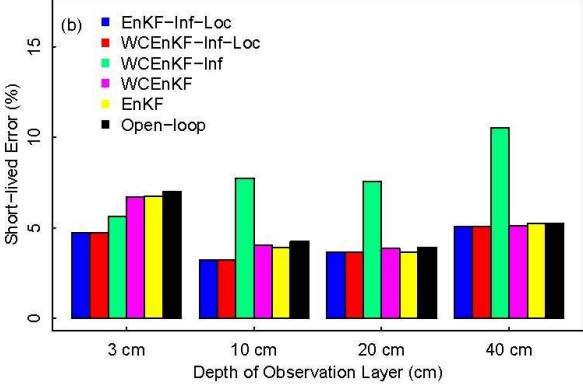

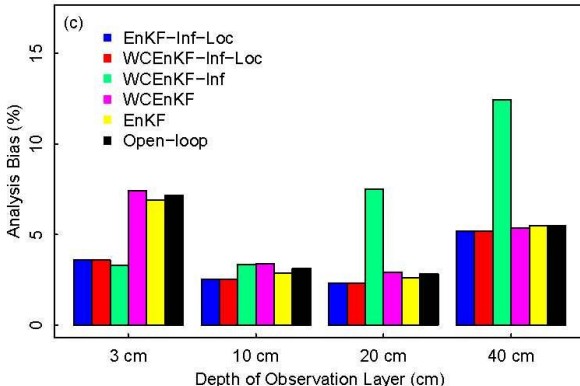

Figure 7. Same as Figure 6, but for BTS station.





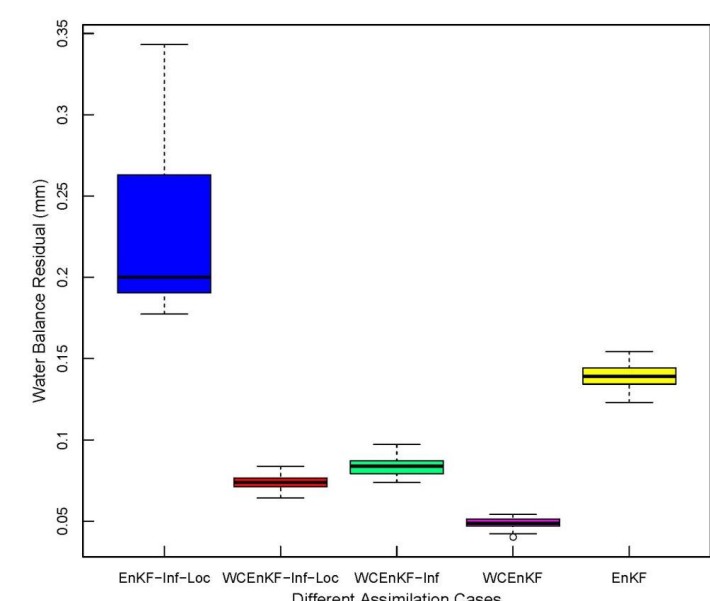

Figure 8. The box plot of the water balance residual in all 40 pixels for the
EnKF-Inf-Loc, WCEnKF-Inf-Loc, WCEnKF-Inf, WCEnKF and EnKF assimilation
schemes.



Table 1. The node depths (cm) of the 10 soil layers in the CoLM model.

| Layer | 1 | 2 | 3 | 4 | 5 | 6 | 7 | 8 | 9 | 10 |
|---|---|---|---|---|---|---|---|---|---|---|
| Depth (cm) | 0.7 | 2.8 | 6.2 | 11.9 | 21.2 | 36.6 | 62.0 | 103.8 | 172.8 | 286.5 |




Table 2. Estimated localization scale factor for different cases.

| Layer | 2 | 3 | 4 | 5 | 6 | 7 | 8 | 9 | 10 |
|---|---|---|---|---|---|---|---|---|---|
| $\mu_s$ | 0.2824 | 0.1256 | 0.0587 | 0.0300 | 0.0163 | 0.0093 | 0.0053 | 0.0025 | 0.0001 |
