# Peer review of "Assimilating Shallow Soil Moisture Observations into Land Models with a Water Budget Constraint"

_Hydrology and Earth System Sciences, 2019_

## Referee Comment (RC1) · Anonymous Referee #1 · 25 Mar 2020

It is with great interest that I read this paper, which investigates a number of different approaches to improve the assimilation of near-surface soil moisture observations. Specifically, the authors use an Ensemble Kalman Filter (EnKF) data assimilation (DA) framework to investigate the effect of forecast error inflation, vertical localization, and a weak water balance constraint. The find that the forecast error inflation helps to reduce the analyses error in upper soil layers that are close to the assimilated observations, however, leads to large analysis errors in deeper soil layers. They conclude that the introduction of a vertical localization function can mitigate the increased analysis error in the deeper soil layers. Finally, the authors concluded that the introduction of a weak water constraint helped to reduce the water residual after the assimilation.

Overall, this is a well-written manuscript that presents a novel and scientifically valuable

contribution to the field of land data assimilation. I have a few minor comments that I would invite the authors to address before the publication of this manuscript.

General comments:

1. One concern I have is the authors' choice not to implement any bias correction. The argument made is that the removing the model bias can lead to issues, but what is lacking is an argument for not implementing the (arguably more traditional) approach of bias correcting the observations to match the model's climatology. By not addressing the bias between the observations and model, you are ultimately violating the assumptions of your DA framework and I would at the very least like to see a discussion on how this impacts the results.

A secondary effect of the lack of bias correction is that it probably aggravates the non-closure of the water balance. While assimilating bias-corrected observations can still lead to a water imbalance, the effect would likely be reduced and requiring less 'intervention' by the water balance constraint introduced in the 'WC' experiments. In this context, it is also worth considering that the nature of observed and modeled soil moisture can be very different (see e.g. Koster et al. 2009), which further necessitates a bias correction step.

Koster, R.D., Guo, Z., Yang, R., Dirmeyer, P.A., Mitchell, K. and Puma, M.J., 2009. On the nature of soil moisture in land surface models. Journal of Climate, 22(16), pp.4322-4335.

2. I would like to see some discussion of the feasibility of the proposed approach for global data assimilation. As the authors discuss, in particular the vertical localization function can be strongly location dependent. I am wondering whether expanding to the global domain would require computing the localization function at each model grid cell and whether that would be computationally feasible? Or would you compute a localization function at a regional scale using soil texture or climate regimes to delineate different regions? Some discussion of the transferability of the presented approach to

global scales would be valuable.

Detailed Comments:

ll. 14-15: This sentence is somewhat redundant, I would recommend rephrasing it. l. 28: 'effectively reduces...' l. 42. Actually, the sub-seasonal to seasonal time scale is probably the one where the land states have the largest impact, so I would mention it here. l. 63: 'land model predictions' l. 67: 'can successfully increase' or 'successfully increases' ll. 65-74: This paragraph focuses a lot on the assimilation of brightness temperatures, while your study is actually investigating methods to improve soil moisture assimilation. So I would include a discussion of soil moisture assimilation here or even a short discussion of soil moisture vs. brightness temperature assimilation. ll. 115-119: Please see general comment on bias correction of observations. ll.128-130: I think you are making an argument here in favor of removing the boas between model and observations. ll.178-182: Just to clarify, the synthetic experiments are conducted over 40 pixels, whereas the real data are conducted only over the two ground station sites, correct? ll. 230-231: I am surprised to see that you are letting the assimilated soil moisture observations update the canopy water content directly, rather than only updating the soil moisture and letting your vegetation module transport the water into the vegetation layer. Is this approach also taken for the real data assimilation experiments, when you are assimilating in situ soil moisture? l. 395: Do you mean 'At each point..'? l. 415: When you say 'the simulation case', do you mean the open loop? ll. 446-449: This paragraph needs some language editing. l. 511: 'threshold'

---

## Referee Comment (RC2) · Anonymous Referee #2 · 2 Apr 2020

The paper titled "Assimilating Shallow Soil Moisture Observations into Land Models with a Water Budget Constraint" presented several modifications to the EnKF data assimilation (DA) that potentially improve DA performance in soil moisture (SM) modeling using shallow-layer observations. A forecast error covariance matrix inflation approach to avoid filter divergence due to underestimated sampling and modeling errors is shown to improve modeling accuracy of SM in layers close to the observation, but leads to increased error in the deeper layers. A vertical localization method is applied to limit the updates to the deep layers to mitigate the errors introduced in the deeper layers. A weak constraint on water balance is able to reduce the water balance residual which is increased due to the forecast error covariance inflation at the price of small increase in the analysis error. Overall the results indicate potential usefulness of such

modifications in improving soil moisture assimilation accuracy of surface soil moisture observations.

However, there is a major issue in the experiment design the raises my concern, i.e. the lack of observation bias-correction. I found the authors' reasoning behind adopting the "traditional bias-blind data assimilation framework" (line 112-117) unconvincing, as there is no evidence to support the "observations" are unbiased relative to the model background in both the synthetic and real-data experiments in this study. Also, it is well-known that remotely sensed soil moisture (the intended application of the proposed modifications) and modeled soil moisture often exhibit different dynamic ranges which warrants the use of a "bias-aware" approach instead (see e.g. Kumar et al. 2012, doi:10.1029/2010WR010261).

In addition, there is an apparent misunderstanding of the Koster et al. (2009) and Reichle and Koster (2004) works where the authors stated that "A major objective of soil moisture data assimilation is to address biases in models and observations" (line 110-111). In fact, both publications indicated the importance of removing the bias in the statistical moments in the observations relative to the model background prior to data assimilation. The major objective of data assimilation is not to remove the bias in model states but to reduce the random, mean-zero noise in the model states, with the model state climatology respected. Even if the observations are considered unbiased, it is recommended that the observations be "scaled" to match the statistical moments of the model states (with long enough time-series). It is well known that directly assimilation of raw observations likely causes model integration to drift, i.e. introduce further bias to the model states. Therefore, the model water balance residual after the soil moisture update in the experiments in this study may be partly attributed to assimilating observations without bias-correction (relative to model), and the true effect of the weak water balance constraint is not accurately revealed.

I would like suggest that the DA experiments repeated with a more robust "bias-aware" approach, to rule out the impact of observation bias in the analysis errors so that the

effects of the proposed modifications are better isolated.

Other minor comments: (line) 65-74: irrelevant to the topic of the paper and should be removed. 229: why directly update canopy water content and snow water equivalent when these two variables are not regulated by near-surface soil moisture? 448: remove "the" following "cover" 478: "the different experiments" –> "different cases" 479-480: for better understanding of the magnitude of improvement, use a percentage scale for water balance residuals 494 "deflation «of» the water balance ..."; "plain" –>"plainly" 511 "shreshould" –>"threshold"

Sec 7.2 Again, one should be careful to use data assimilation to achieve "bias correction" in model states. This is another example of misunderstanding the major objective of DA in this work. Seemingly reduced systematic bias in modeled soil moisture may be an artifact due to biased observation relative to model background.

---

## Author Comment (AC1) · 17 May 2020

**Journal:** Hydrology and Earth System Sciences

**Title:** Assimilating Shallow Soil Moisture Observations into Land Models with a Water Budget Constraint

**Authors:** Bo Dan, Xiaogu Zheng, Guocan Wu, and Tao Li

**MS NO.:** hess-2019-696

**MS Type:** Research Article

The authors highly appreciate the anonymous reviewer for his/her very helpful and insightful comments that lead to the considerable improvement of the quality of this manuscript. We have checked our work carefully according to these comments and made the requested changes.

Below we indicate the comments and use blue font for our responses. The corresponding revised texts are also used blue font in the revised version of our manuscript.

**Reviewer #1**

It is with great interest that I read this paper, which investigates a number of different approaches to improve the assimilation of near-surface soil moisture observations. Specifically, the authors use an Ensemble Kalman Filter (EnKF) data assimilation (DA) framework to investigate the effect of forecast error inflation, vertical localization, and a weak water balance constraint. The find that the forecast error inflation helps to reduce the analyses error in upper soil layers that are close to the assimilated observations, however, leads to large analysis errors in deeper soil layers. They conclude that the introduction of a vertical localization function can mitigate the increased analysis error in the deeper soil layers. Finally, the authors concluded that the introduction of a weak water constraint helped to reduce the water residual after the assimilation.

Overall, this is a well-written manuscript that presents a novel and scientifically valuable contribution to the field of land data assimilation. I have a few minor comments that I would invite the authors to address before the publication of this manuscript.

**Response:** Thank you very much for your thorough reviewing and valuable comments.

**General comments:**

1. One concern I have is the authors' choice not to implement any bias correction. The argument made is that the removing the model bias can lead to issues, but what is lacking is an argument for not implementing the (arguably more traditional) approach of bias correcting the observations to match the model's climatology. By not

addressing the bias between the observations and model, you are ultimately violating the assumptions of your DA framework and I would at the very least like to see a discussion on how this impacts the results.

**Response:** Thank you for your comments. Following it and the major comment of another reviewer, the bias-aware data assimilation proposed by Dee (2000) was applied to further correct the bias of the analysis states assimilated using WCEnKF-Inf-Loc. This scheme was named as WCEnKF-Inf-Loc-BA, and the corresponding results were added in Figures 5-6.

Figure 5 shows that, the spatial averaged root analysis error variances of WCEnKF-Inf-Loc and WCEnKF-Inf-Loc-BA were comparable (2.12% for the WCEnKF-Inf-Loc-BA and 2.16% for the WCEnKF-Inf-Loc) for the layers that are shallower than 36.6 cm. This could be due to that the observations are closer to the shallow layers and the vertical localization approach is reasonably effective to reduce the bias. However, for the layers that are deeper than 62.0 cm, the averaged root analysis error of the WCEnKF-Inf-Loc-BA (6.05%) was less than that of the WCEnKF-Inf-Loc (6.59%). This indicates that the bias correction is useful for this experiment, especially for the soil moistures in deeper layers. (Lines 420-428)

[Figure]

Figure 5. The assimilation results in each layer for the five schemes: a weakly constrained bias-aware ensemble Kalman filter with forecast error inflation and vertical localization (WCEnKF-Inf-Loc-BA), a weakly constrained ensemble Kalman filter with forecast error inflation and vertical localization (WCEnKF-Inf-Loc), a weakly constrained ensemble Kalman filter with forecast error inflation (WCEnKF-Inf), a weakly constrained ensemble Kalman filter (WCEnKF), and the traditional assimilation (EnKF). Graphic (a) is for spatial averaged analysis error of the soil moisture content, (b) is for the short-lived error and (c) is for the analysis bias.

A secondary effect of the lack of bias correction is that it probably aggravates the non-closure of the water balance. While assimilating bias-corrected observations can

still lead to a water imbalance, the effect would likely be reduced and requiring less 'intervention' by the water balance constraint introduced in the 'WC' experiments. In this context, it is also worth considering that the nature of observed and modeled soil moisture can be very different (see e.g. Koster et al. 2009), which further necessitates a bias correction step.

Koster, R.D., Guo, Z., Yang, R., Dirmeyer, P.A., Mitchell, K. and Puma, M.J., 2009. On the nature of soil moisture in land surface models. Journal of Climate, 22(16), pp.4322-4335.

**Response:** Thank you for your comments. In the revised version, the water budget residuals of different assimilation schemes were shown in Figure 6. The spatial average of the water balance residuals for WCEnKF-Inf-Loc-BA scheme was 0.0723 mm, which was slightly smaller than that for WCEnKF-Inf-Loc scheme (0.0737 mm). The small improvement on water balance residuals may be due to the small improvement on analysis bias by the additional bias-aware assimilation, but it suggests a tendency of the bias correction to further reduce the water balance budget. (Lines 442-445)

[Figure]

Figure 6. The box plot of the water balance residual in all 40 pixels for the WCEnKF-Inf-Loc-BA, WCEnKF-Inf-Loc,WCEnKF-Inf, WCEnKF and EnKF assimilation schemes.

2. I would like to see some discussion of the feasibility of the proposed approach for global data assimilation. As the authors discuss, in particular the vertical localization function can be strongly location dependent. I am wondering whether expanding to the global domain would require computing the localization function at each model grid cell and whether that would be computationally feasible? Or would you compute a localization function at a regional scale using soil texture or climate regimes to delineate different regions? Some discussion of the transferability of the presented approach to global scales would be valuable.

**Response:** Thank you for your suggestions. As you pointed out, the most computational cost in the assimilation system is on computing the localization function at each model grid cell. For the synthetic experiments with CoLM model and 40 grids, it takes about 24 hours running on the personal workstation. For global data assimilation with $2^o$ resolution it could take about 3 months to finish if running on the personal workstation, but the super server and parallel computation can significantly shorten the computational time. We also agree with you that "a regional scale using soil texture or climate regimes can be used to delineate different regions". By this way, the computational time of global data assimilation can be further reduced. (Lines 525-532)

**Detailed Comments:**

ll. 14-15: This sentence is somewhat redundant, I would recommend rephrasing it.

**Response:** We have revised this sentence to "Assimilating observations of shallow soil moisture content into land models is an important step in estimating soil moisture content." (Line 14-15)

l. 28: 'effectively reduces. . .'

**Response:** Revised.

l. 42. Actually, the sub-seasonal to seasonal time scale is probably the one where the land states have the largest impact, so I would mention it here.

**Response:** We have added "at sub-seasonal to seasonal time scale".

l. 63: 'land model predictions'

**Response:** Revised.

l. 67: 'can successfully increase' or 'successfully increases'

**Response:** The sentence has been deleted following the next comment.

ll. 65-74: This paragraph focuses a lot on the assimilation of brightness temperatures, while your study is actually investigating methods to improve soil moisture assimilation. So I would include a discussion of soil moisture assimilation here or even a short discussion of soil moisture vs. brightness temperature assimilation.

**Response:** Thanks for your comment. This paragraph has been revised as follows. (Lines 60-69)

Many studies indicated that a better approach to improving the estimates of soil moisture contents on regional scales is to constrain land model predictions by

assimilating surface soil moisture data (Crow and Loon 2006; Crow and Wood 2003; Reichle and Koster 2005). It can provide better estimates of the true soil moisture content column states than the model forecasts (Crow *et al*. 2017; Lu et al. 2012; Lu *et al*. 2015), and can further improve land surface model initial conditions for coupled short-term weather prediction (Chen *et al*. 2014; Santanello *et al*. 2016; Yang *et al*. 2016). Especially, surface soil moisture data can be provided by in-situ observations and passive microwave measurements (brightness temperatures) observed by remote sensing.

ll. 115-119: Please see general comment on bias correction of observations.

**Response:** Thanks for your comment. In the revised version, the sentence "Therefore in this study, we still use traditional bias-blind data assimilation framework." was replaced as "However, bias can be detected by monitoring observation-minus-forecast statistics in the assimilation system (Dee and Todling 2000). Then a bias-aware assimilation method can be designed to estimate and correct the systematic errors sequentially with the model state variables (Dee 2005). Such bias correction method is adopted in this study to detect the performance among different assimilation schemes." (Lines 109-114)

ll.128-130: I think you are making an argument here in favor of removing the bias between model and observations.

**Response:** Yes, we have added the bias correction schemes in the revised version.

ll.178-182: Just to clarify, the synthetic experiments are conducted over 40 pixels, whereas the real data are conducted only over the two ground station sites, correct?

**Response:** Yes.

ll. 230-231: I am surprised to see that you are letting the assimilated soil moisture observations update the canopy water content directly, rather than only updating the soil moisture and letting your vegetation module transport the water into the vegetation layer. Is this approach also taken for the real data assimilation experiments, when you are assimilating in situ soil moisture?

**Response:** We agree that not update the canopy water content is an option. The approach in this study is adopted from Yilmaz *et al*. (2011; 2012) where the canopy water content was updated. This approach was also taken when we are assimilating in situ soil moistures at the two Mongolia stations.

l. 395: Do you mean 'At each point..'?

**Response:** Yes, the words have been revised.

l. 415: When you say 'the simulation case', do you mean the open-loop?

**Response:** Yes it is open-loop. However, the results for the open-loop are not shown in Figures 5 and 6 in the revised version.

ll. 446-449: This paragraph needs some language editing.

**Response:** We have removed the observation study in the revised version. This is because the paper is too long after adding the bias-aware data assimilation experiment, so we would like to shorten it. Also the observations are not available in all layers, and then it is more difficult to validate the proposed assimilation methods. Since these paragraphs were related to observation study, they were deleted in the revised version.

l. 511: 'threshold'

**Response:** The typo has been revised.

Again, thank you very much for your thorough reviewing and valuable comments. The references in this reply are listed as follows, while some of them have already in the previous version of the manuscript.

Chen, F., Crow, W.T. and Ryu, D., 2014. Dual Forcing and State Correction via Soil Moisture Assimilation for Improved Rainfall-Runoff Modeling. *Journal of Hydrometeorology*, 15(5): 1832-1848.

Crow, W.T., Chen, F., Reichle, R.H. and Liu, Q., 2017. L band microwave remote sensing and land data assimilation improve the representation of prestorm soil moisture conditions for hydrologic forecasting. *Geophysical Research Letters*, 44(11): 5495-5503.

Crow, W.T. and Loon, E.V., 2006. Impact of incorrect model error assumptions on the sequential assimilation of remotely sensed surface soil moisture. *Journal of Hydrometeorology*, 7: 421-432.

Crow, W.T. and Wood, E.F., 2003. The assimilation of remotely sensed soil brightness temperature imagery into a land surface model using Ensemble Kalman filtering: a case study based on ESTAR measurements during SGP97. *Advances in Water Resources*, 26: 137-149.

Dee, D.P., 2005. Bias and data assimilation. *Quarterly Journal of the Royal*

*Meteorological Society*, 131: 3323-3343.

Dee, D.P. and Todling, R., 2000. Data assimilation in the presence of forecast bias: The GEOS moisture analysis. *Monthly Weather Review*, 128(9): 3268-3282.

Lu, H., Koike, T., Yang, K., Hu, Z.Y., Xu, X.D., Rasmy, M., Kuria, D. and Tamagawa, K., 2012. Improving land surface soil moisture and energy flux simulations over the Tibetan plateau by the assimilation of the microwave remote sensing data and the GCM output into a land surface model. *International Journal of Applied Earth Observation and Geoinformation*, 17: 43-54.

Lu, H., Yang, K., Koike, T., Zhao, L. and Qin, J., 2015. An Improvement of the Radiative Transfer Model Component of a Land Data Assimilation System and Its Validation on Different Land Characteristics. *Remote Sensing*, 7(5): 6358-6379.

Reichle, R.H. and Koster, R.D., 2005. Global assimilation of satellite surface soil moisture retrievals into the NASA Catchment land surface model. *Geophysical Reasearch Letters*, 32.

Santanello, J.A., Kumar, S.V., Peters-Lidard, C.D. and Lawston, P.M., 2016. Impact of Soil Moisture Assimilation on Land Surface Model Spinup and Coupled Land-Atmosphere Prediction. *Journal of Hydrometeorology*, 17(2): 517-540.

Yang, K., Zhu, L., Chen, Y., Zhao, L., Qin, J., Lu, H., Tang, W., Han, M., Ding, B. and Fang, N., 2016. Land surface model calibration through microwave data assimilation for improving soil moisture simulations. *Journal of Hydrology*,

533: 266-276.

Yilmaz, M.T., Delsole, T. and Houser, P.R., 2011. Improving land data assimilation performance with a water budget constraint. *Journal of Hydrometeorology*, 12: 1040-1055.

Yilmaz, T.M., Delsole, T. and Houser, P.R., 2012. Reducing water imbalance in land data assimilation: Ensemble filtering without perturbed observations. *Journal of Hydrometeorology*, 13(1): 413-420.

---

## Author Comment (AC2) · 17 May 2020

**Journal:** Hydrology and Earth System Sciences

**Title:** Assimilating Shallow Soil Moisture Observations into Land Models with a Water Budget Constraint

**Authors:** Bo Dan, Xiaogu Zheng, Guocan Wu, and Tao Li

**MS NO.:** hess-2019-696

**MS Type:** Research Article

The authors highly appreciate the anonymous reviewer for his/her very helpful and insightful comments that lead to the considerable improvement of the quality of this manuscript. We have checked our work carefully according to these comments and made the requested changes.

Below we indicate the comments and use blue font for our responses. The corresponding revised texts are also used blue font in the revised version of our manuscript.

**Reviewer #2**

The paper titled "Assimilating Shallow Soil Moisture Observations into Land Models with a Water Budget Constraint" presented several modifications to the EnKF data assimilation (DA) that potentially improve DA performance in soil moisture (SM) modeling using shallow-layer observations. A forecast error covariance matrix inflation approach to avoid filter divergence due to underestimated sampling and modeling errors is shown to improve modeling accuracy of SM in layers close to the observation, but leads to increased error in the deeper layers. A vertical localization method is applied to limit the updates to the deep layers to mitigate the errors introduced in the deeper layers. A weak constraint on water balance is able to reduce the water balance residual which is increased due to the forecast error covariance inflation at the price of small increase in the analysis error. Overall the results indicate potential usefulness of such modifications in improving soil moisture assimilation accuracy of surface soil moisture observations.

**Response:** Thank you very much for your thorough reviewing and valuable comments.

However, there is a major issue in the experiment design the raises my concern, i.e. the lack of observation bias-correction. I found the authors' reasoning behind adopting the "traditional bias-blind data assimilation framework" (line 112-117) unconvincing, as there is no evidence to support the "observations" are unbiased relative to the model background in both the synthetic and real-data experiments in this study. Also, it is well known that remotely sensed soil moisture (the intended application of the proposed modifications) and modeled soil moisture often exhibit different dynamic ranges which warrants the use of a "bias-aware" approach instead

(see e.g. Kumar et al. 2012, doi:10.1029/2010WR010261).

**Response:** Thank you for your comment. Following it and the major comment of the other reviewer, the bias-aware data assimilation proposed by Dee (2005) was applied to further correct the bias of the analysis states assimilated using WCEnKF-Inf-Loc. This scheme was named as WCEnKF-Inf-Loc-BA, and the corresponding results were added in Figures 5-6.

Figure 5 shows that, the spatial averaged root analysis error variances of WCEnKF-Inf-Loc and WCEnKF-Inf-Loc-BA were comparable (2.12% for the WCEnKF-Inf-Loc-BA and 2.16% for the WCEnKF-Inf-Loc) for the layers that are shallower than 36.6 cm. However, for the layers that are deeper than 62.0 cm, the averaged root analysis error of the WCEnKF-Inf-Loc-BA (6.05%) was less than that of the WCEnKF-Inf-Loc (6.59%). This indicated that the bias correction is useful for this experiment, especially for the soil moistures in deeper layers. (Lines 420-428)

[Figure]

Figure 5. The assimilation results in each layer for the five schemes: a weakly constrained bias-aware ensemble Kalman filter with forecast error inflation and vertical localization (WCEnKF-Inf-Loc-BA), a weakly constrained ensemble Kalman filter with forecast error inflation and vertical localization (WCEnKF-Inf-Loc), a weakly constrained ensemble Kalman filter with forecast error inflation (WCEnKF-Inf), a weakly constrained ensemble Kalman filter (WCEnKF), and the traditional assimilation (EnKF). Graphic (a) is for spatial averaged analysis error of the soil moisture content, (b) is for the short-lived error and (c) is for the analysis bias.

In addition, there is an apparent misunderstanding of the Koster et al. (2009) and Reichle and Koster (2004) works where the authors stated that "A major objective of

soil moisture data assimilation is to address biases in models and observations" (line 110-111). In fact, both publications indicated the importance of removing the bias in the statistical moments in the observations relative to the model background prior to data assimilation. The major objective of data assimilation is not to remove the bias in model states but to reduce the random, mean-zero noise in the model states, with the model state climatology respected. Even if the observations are considered unbiased, it is recommended that the observations be "scaled" to match the statistical moments of the model states (with long enough time-series). It is well known that directly assimilation of raw observations likely causes model integration to drift, i.e. introduce further bias to the model states. Therefore, the model water balance residual after the soil moisture update in the experiments in this study may be partly attributed to assimilating observations without bias-correction (relative to model), and the true effect of the weak water balance constraint is not accurately revealed.

I would like suggest that the DA experiments repeated with a more robust "bias-aware" approach, to rule out the impact of observation bias in the analysis errors so that the effects of the proposed modifications are better isolated.

**Response:** Thank you for your comments. We agree that the model state climatology should be respected and directly assimilation of raw observations likely causes model integration to drift. We also agree that a more robust "bias-aware" approach is necessary, because it respects the model state climatology and uses the estimated bias to prevent model integration to drift. In the revised version, the bias-aware data assimilation proposed by Dee (2005) was investigated.

In the revised version, the water budget residuals of different assimilation schemes were shown in Figure 6. The spatial average of the water balance residuals for WCEnKF-Inf-Loc-BA scheme was 0.0723 mm, which was slightly smaller than

that for WCEnKF-Inf-Loc scheme (0.0737 mm). The small improvement on water balance residuals may be due to the small improvement on analysis bias by the additional bias-aware assimilation, but it suggests a tendency of the bias correction to further reduce the water balance budget. (Lines 442-445)

[Figure]

Figure 6. The box plot of the water balance residual in all 40 pixels for the WCEnKF-Inf-Loc-BA, WCEnKF-Inf-Loc,WCEnKF-Inf, WCEnKF and EnKF assimilation schemes.

**Other minor comments:**

Lines 65-74: irrelevant to the topic of the paper and should be removed.

**Response:** Thanks for the comment. Following it and the comment of the other reviewer, lines 65-74 were removed and the paragraph is revised as follows:

Many studies indicated that a better approach to improving the estimates of soil moisture contents on regional scales is to constrain land model predictions by assimilating surface soil moisture data (Crow and Loon 2006; Crow and Wood 2003; Reichle and Koster 2005). It can provide better estimates of the true soil moisture

content column states than the model forecasts (Crow *et al*. 2017; Lu *et al*. 2012; Lu *et al*. 2015), and can further improve land surface model initial conditions for coupled short-term weather prediction (Chen *et al*. 2014; Santanello *et al*. 2016; Yang *et al*. 2016). Especially, surface soil moisture data can be provided by in-situ observations and passive microwave measurements (brightness temperatures) observed by remote sensing. (Lines 60-69)

Line 229: why directly update canopy water content and snow water equivalent when these two variables are not regulated by near-surface soil moisture?

**Response:** We agree that not update the canopy water content and snow water equivalent is an option. The approach in this study is adopted from Yilmaz *et al*. (2011; 2012) where the canopy water content and snow water equivalent were updated.

Line 448: remove "the" following "cover"

**Response:** Revised.

Line 478: "the different experiments" –> "different cases"

**Response:** This paragraph was related to cases of the DGS and BTS stations, and was removed in the revised version.

Lines 479-480: for better understanding of the magnitude of improvement, use a percentage scale for water balance residuals

**Response:** Thank you for your suggestion. In the revised version, the real data experiments were deleted due to the length of the manuscript.

Line 494 "deflation ⟨of⟩ the water balance ..."; "plain" –>"plainly"

**Response:** Revised.

Line 511 "shreshould" –>"threshold"

**Response:** Revised.

Sec 7.2 Again, one should be careful to use data assimilation to achieve "bias correction" in model states. This is another example of misunderstanding the major objective of DA in this work. Seemingly reduced systematic bias in modeled soil moisture may be an artifact due to biased observation relative to model background.

**Response:** Thank you for your comment. We have added the experimentation with bias correction method. The results shows that the bias-ware assimilation schemes can further reduce the analysis error and water budget residuals.

Again, thank you very much for your thorough reviewing and valuable comments. The references in this reply are listed as follows, while some of them have already in the previous version of the manuscript.

Chen, F., Crow, W.T. and Ryu, D., 2014. Dual Forcing and State Correction via Soil Moisture Assimilation for Improved Rainfall-Runoff Modeling. *Journal of Hydrometeorology*, 15(5): 1832-1848.

Crow, W.T., Chen, F., Reichle, R.H. and Liu, Q., 2017. L band microwave remote sensing and land data assimilation improve the representation of prestorm soil moisture conditions for hydrologic forecasting. *Geophysical Research Letters*,

44(11): 5495-5503.

Crow, W.T. and Loon, E.V., 2006. Impact of incorrect model error assumptions on the sequential assimilation of remotely sensed surface soil moisture. *Journal of Hydrometeorology*, 7: 421-432.

Crow, W.T. and Wood, E.F., 2003. The assimilation of remotely sensed soil brightness temperature imagery into a land surface model using Ensemble Kalman filtering: a case study based on ESTAR measurements during SGP97. *Advances in Water Resources*, 26: 137-149.

Lu, H., Koike, T., Yang, K., Hu, Z.Y., Xu, X.D., Rasmy, M., Kuria, D. and Tamagawa, K., 2012. Improving land surface soil moisture and energy flux simulations over the Tibetan plateau by the assimilation of the microwave remote sensing data and the GCM output into a land surface model. *International Journal of Applied Earth Observation and Geoinformation*, 17: 43-54.

Lu, H., Yang, K., Koike, T., Zhao, L. and Qin, J., 2015. An Improvement of the Radiative Transfer Model Component of a Land Data Assimilation System and Its Validation on Different Land Characteristics. *Remote Sensing*, 7(5): 6358-6379.

Reichle, R.H. and Koster, R.D., 2005. Global assimilation of satellite surface soil moisture retrievals into the NASA Catchment land surface model. *Geophysical Reasearch Letters*, 32.

Santanello, J.A., Kumar, S.V., Peters-Lidard, C.D. and Lawston, P.M., 2016. Impact of Soil Moisture Assimilation on Land Surface Model Spinup and Coupled

Land-Atmosphere Prediction. *Journal of Hydrometeorology*, 17(2): 517-540.

Yang, K., Zhu, L., Chen, Y., Zhao, L., Qin, J., Lu, H., Tang, W., Han, M., Ding, B. and Fang, N., 2016. Land surface model calibration through microwave data assimilation for improving soil moisture simulations. *Journal of Hydrology*, 533: 266-276.

Yilmaz, M.T., Delsole, T. and Houser, P.R., 2011. Improving land data assimilation performance with a water budget constraint. *Journal of Hydrometeorology*, 12: 1040-1055.

Yilmaz, T.M., Delsole, T. and Houser, P.R., 2012. Reducing water imbalance in land data assimilation: Ensemble filtering without perturbed observations. *Journal of Hydrometeorology*, 13(1): 413-420.

---

## Author Response (AR2)

**Journal:** Hydrology and Earth System Sciences

**Title:** Assimilating Shallow Soil Moisture Observations into Land Models with a Water Budget Constraint

**Authors:** Bo Dan, Xiaogu Zheng, Guocan Wu, and Tao Li

**MS NO.:** hess-2019-696

**MS Type:** Research Article

The authors highly appreciate the anonymous reviewer for his/her very helpful and insightful comments that lead to the considerable improvement of the quality of this manuscript. We have checked our work carefully according to these comments and made the requested changes. The main improvement is the discussions on updating the canopy water content and WCEnKF in reducing water budget residual. The Abstract and Conclusions sections are also revised with adding necessary quantitive measures.

Below we indicate the comments and use blue font for our responses. The corresponding revised texts are also used blue font in the revised version of our manuscript.

The study is suitable for publication after following minor changes.

(1) Both reviewers the prior version of the manuscript asked about the direct update of canopy water content. I understand that the authors are following the approach of Yilmaz et al. (2011; 2012) but this choice of updating canopy water content needs to be discussed further in the manuscript. It could be done in the discussion section or in the methods section.

**Response:** Thanks for your comment. The canopy's water content (CWC) and snow water equivalent (SWE) are related to the water budget. If the water budget constraint is absent, they are normally not updated and the vegetation module transports the water into the vegetation layer. However, the present study focused on the assimilation with the water budget constraint, then updating CWC and SWE would help to reduce the water budget residuals.

For the assimilation with the water budget constraint but without update of CWC and SWE, the state variables related to the water budget are decomposed as $\mathbf{x} = \left( {}_1\mathbf{x}, {}_2\mathbf{x} \right)$ where ${}_1\mathbf{x}$ comprises of SM and SIC (the soil moisture content and the soil ice content at the 10 vertical levels listed in Table 1), ${}_2\mathbf{x}$ comprises of CWC and SWE (the canopy's water content and the snow water equivalent). $\mathbf{c} = \left( {}_1\mathbf{c}, {}_2\mathbf{c} \right)$ is a 22-dimensional vector that converts the units of $\mathbf{x} = \left( {}_1\mathbf{x}, {}_2\mathbf{x} \right)$ to millimeters (mm). The assimilation for not update of ${}_2\mathbf{x}$ can be achieved by substituting $\mathbf{x}$ and $\beta_{n,t}$ in section 3.2 by ${}_1\mathbf{x}$ and ${}_1\beta_{n,t}$ respectively, that is

$$ {}_1\beta_{n,t} = {}_1\mathbf{c}^{\mathrm{T}} {}_1\mathbf{x}^a_{n,t-1} + {}_2\mathbf{c}^{\mathrm{T}} {}_2\mathbf{x}^f_{n,t-1} + Pr_t - Ev^f_{n,t} - Rn^f_{n,t}, \tag{22} $$

where $Pr_t$, $Ev^f_{n,t}$ and $Rn^f_{n,t}$ are diagnostic variables specifying the states of the precipitation, evapotranspiration and runoff, respectively. By this way, the canopy's water content are not updated and the vegetation module transports the water into the vegetation layer. In this study, the range of the estimated CWCs for all assimilations with or without update of ${}_2\mathbf{x}$ is only about 0.005 mm. Considering the estimated water budget residuals are between 0.05 mm and 0.14 mm and there is no SWE in the summer peried, we conclude that update of CWC has a little impact on water balance in this study.

This discussion was added in section 6.3 of the revised version. (Lines 539-561)

(2) The results in Fig. 6 indicate that WcEnKF results in the smallest water balance residual relative to various WcEnKF-Inf and WcEnKF-Inf-Loc. I realize that WcEnKF-Inf and WcEnKF-Inf-Loc leads to smaller bias in soil moisture but if the focus of a study or experiment is reducing water balance, does this result indicate that WcEnKF is a better choice? I assume it is computationally faster to implement WcEnKF too. Please discuss this point.

**Response:** Thanks for your comment. We agree that if the focus of a study or experiment is reducing water balance, WCEnKF could be a better choice and computationally faster than WCEnKF-Inf and WCEnKF-Inf-Loc schemes. Accordingly, it is plainly obvious that the water balance residual of the scheme WCEnKF-Inf is larger than that of the scheme WCEnKF. However, the objective in this study is to reduce water balance without significantly increasing the analysis error. Since the analysis errors for WCEnKF in the layers shallower than 36.6 cm are significantly larger than those for the schemes with inflation, WCEnKF is not preferred.

These texts have been added to the revised version. (Lines 468-476)

(3) Both Abstract and Conclusions section do not mention any quantitive measures (e.g. % improvement in bias) of improvement in model performance after data assimilation.

**Response:** Thanks for your comment. The main quantitive measures of the analysis errors and water budget residuals are included in the abstract and conclusions.

For the more details, in abstract we added (Lines 27-35):

"The results of the assimilation process suggest that the inflation approach effectively reduces the analysis error from 6.70% to 2.00% in shallow layers, but increases from 6.38% to 12.49% in deep layers. The vertical localization approach leads to 6.59% of the analysis error in deep layers, and the bias-aware assimilation scheme further reduces to 6.05% . The spatial average of the water balance residual is 0.0487 mm of weakly constrained EnKF scheme, and 0.0737 mm of weakly constrained EnKF with inflation and localization scheme, which are much smaller than 0.1389 mm of the EnKF scheme."

In the conclusion we added (Lines 589-593):

"The experiment results of synthetic study show that the WCEnKF-Inf-Loc assimilation scheme can reduce the analysis error from 6.70% to 2.00% in the shallow layers, with both the short-lived analysis error and the analysis bias reduced. It also leads to a rational water budget residual with spatial average 0.0737 mm, which is much smaller than 0.1389 mm of the EnKF scheme."

(4) Line 29: "Finnaly" should be "Finally".
**Response:** Revised.

(5) Line 78: "it suggests" should be "it is suggested".
**Response:** Revised.

(6) 4.2.3, please define the variables in the equation to calculate water balance residuals.
**Response:** The variables are defined as follows: "$N$ is the ensemble size, $a_{ts}$ is the number of assimilation time steps, and $r_{n,t}$ is the ensemble water budget residual at time step t as defined in Eq. (6)."
    This was added in section 4.2.3. (Lines 361-362)

(7): 6.3 "Notes" is a pretty vague heading for this section, perhaps "Broader implications" or "Global implementation" would be better.
**Response:** The heading has been changed to "Broader implications".

Again, thanks for your valuable comments and recommendation.

The main changes are listed as follows.

(1) Lines 27-35: Added the main quantitive measures of the analysis errors and water budget residuals in the abstract.

(2) Lines 361-362: Added the variables in the equation to calculate water balance residuals (Eq. (21)).

(3) Lines 468-476: Added the discussion on WCEnKF scheme.

(4) Lines 539-561: Added the discussion on updating of canopy water content.

(5) Lines 589-593: Added the main quantitive measures of the analysis errors and water budget residuals in the conclusions.

[revised manuscript text omitted]
$ | 0.2824 | 0.1256 | 0.0587 | 0.0300 | 0.0163 | 0.0093 | 0.0053 | 0.0025 | 0.0001 |